# Thiophene-Linked 1,2,4-Triazoles: Synthesis, Structural Insights and Antimicrobial and Chemotherapeutic Profiles [note 1]

**DOI:** 10.3390/ph17091123

**Published:** 2024-08-25

**Authors:** Nada A. El-Emam, Mahmoud B. El-Ashmawy, Ahmed A. B. Mohamed, El-Sayed E. Habib, Subbiah Thamotharan, Mohammed S. M. Abdelbaky, Santiago Garcia-Granda, Mohamed A. A. Moustafa

**Affiliations:** 1Department of Medicinal Chemistry, Faculty of Pharmacy, Mansoura University, Mansoura 35516, Egypt; nada_el_emam@hotmail.com (N.A.E.-E.); ahmed_smt@yahoo.com (A.A.B.M.); maastafa0828@gmail.com (M.A.A.M.); 2Department of Microbiology and Immunology, Faculty of Pharmacy, Mansoura University, Mansoura 35516, Egypt; sayedhabib@mans.edu.eg; 3Biomolecular Crystallography Laboratory and DBT-Bioinformatics Center, School of Chemical and Biotechnology, SASTRA Deemed University, Thanjavur 613 401, India; thamu@scbt.sastra.edu; 4Department of Physical Chemistry, Faculty of Chemical Sciences, University of Salamanca, 37008 Salamanca, Spain; mohammed@usal.es; 5Department of Physical and Analytical Chemistry, Faculty of Chemistry, University of Oviedo-CINN (CSIC), 33006 Oviedo, Spain

**Keywords:** 1,2,4-triazoles, thiophene, antimicrobial activity, anti-proliferative activity, molecular docking, single-crystal XRD

## Abstract

The reaction of thiophene-2-carbohydrazide **1** or 5-bromothiophene-2-carbohydrazide **2** with various haloaryl isothiocyanates and subsequent cyclization by heating in aqueous sodium hydroxide yielded the corresponding 4-haloaryl-5-(thiophen-2-yl or 5-bromothiophen-2-yl)-2,4-dihydro-3*H*-1,2,4-triazole-3-thione **5a**-**e**. The triazole derivatives **5a** and **5b** were reacted with different secondary amines and formaldehyde solution to yield the corresponding 2-aminomethyl-4-haloaryl-2,4-dihydro-3*H*-1,2,4-triazole-3-thiones **6a**–**e**, **7a**–**e**, **8**, **9**, **10a** and **10b** in good yields. The in vitro antimicrobial activity of compounds **5a**–**e**, **6a**–**e**, **7a**–**d**, **8**, **9**, **10a** and **10b** was evaluated against a panel of standard pathogenic bacterial and fungal strains. Compounds **5a**, **5b**, **5e**, **5f**, **6a**–**e**, **7a**–**d**, **8**, **9**, **10a** and **10b** showed marked activity, particularly against the tested Gram-positive bacteria and the Gram-negative bacteria *Escherichia coli*, and all the tested compounds were almost inactive against all the tested fungal strains. In addition, compounds **5e**, **6a**–**e**, **7a**–**d** and **10a** exhibited potent anti-proliferative activity, particularly against HepG-2 and MCF-7 cancer cell lines (IC_50_ < 25 μM). A detailed structural insight study based on the single crystals of compounds **5a**, **5b**, **6a**, **6d** and **10a** is also reported. Molecular docking studies of the highly active antibacterial compounds **5e**, **6b**, **6d**, **7a** and **7d** showed a high affinity for DNA gyrase. Meanwhile, the potent anti-proliferative activity of compounds **6d**, **6e** and **7d** may be attributed to their high affinity for cyclin-dependent kinase 2 (CDK2).

## 1. Introduction

1,2,4-Triazole heterocycle was early identified as the crucial core of numerous therapeutically interesting drugs with a wide spectrum of chemotherapeutic activities [1,2]. Triazole-based drugs are widely used as a useful medication for the treatment of topical and systemic fungal diseases [3,4]. Fluconazole [5], itraconazole [6], voriconazole [7] and posaconazole [8] are among the currently used antifungal agents. In addition, 1,2,4-triazole-based derivatives were reported to endow potent anticancer activity [9,10]. Bemcentinib [11], letrozole [12], vorozole [13], and anastrazole [14] are currently used as an efficient treatment for different cancers. In addition, the specific tankyrase inhibitor G007-LK was recently approved for clinical trials for the treatment of breast and colorectal cancers [15] (Figure 1).

In recent years, some novel 1,2,4-triazoles have been reported to demonstrate marked antibacterial [16,17,18] and anti-tuberculosis activities [19]. Furthermore, ribavirin [20] and other related 1,2,4-triazole derivatives [21] were recognized as potent antiviral agents (Figure 1).

On the other hand, thiophene heterocycle constitutes a major building block of many drugs [22,23,24]. The thiophene–triazole hybrid analog PF-4989216, a potent and selective PI3K kinase inhibitor, was discovered as an effective anticancer agent [25]. OSI-930 is an orally active anticancer agent acting by inhibitors of c-Kit and VEGFR-2, and it shows broad efficacy in tumor models representative of small cell lung cancer, glioblastoma, colorectal, renal, head and neck, non-small cell lung cancer and gastric cancers [26,27]. MCL0527 was identified as a potent anti-proliferative agent via p53-MDM2 binding inhibition [28]. The thiophene-based antifungal drugs tioconazole [29] and sertaconazole [30] are currently used for the treatment of fungal skin infections. The non-nucleoside polymerase inhibitor lomibuvir (VX-222) is currently used for the treatment of chronic hepatitis C virus (HCV) infections [31] (Figure 2).

In view of the aforementioned findings, we describe herein the synthesis, characterization, preliminary antimicrobial and anti-proliferative activities of a series of thiophene-linked 1,2,4-triazole derivatives. In addition, the single-crystal X-ray structures of five representative compounds were studied to investigate various intermolecular interactions including N–H···S hydrogen bond and C–H···S/N/F/π interactions. Additionally, the σ-hole interactions such as halogen (Br···F) and chalcogen bonds (S···S/π) [32,33,34] were also investigated.

## 2. Results and Discussion

### 2.1. Chemical Synthesis

Thiophene-2-carbohydrazide **3a** and 5-bromothiophene-2-carbohydrazide **3b** were prepared starting with their corresponding carboxylic acids **1a** and **1b** via esterification to their corresponding esters **2a** and **2b**, and subsequent hydrazinolysis following the previously reported procedures [35,36]. Treatment of the carbohydrazides **3a** and **3b** with the corresponding haloaryl isothiocyanate by heating in ethanol yielded the intermediate *N*-aryl-2-(thiophene-2-carbonyl or 5-bromothiophene-2-carbonyl)hydrazine-1-carbothioamides **4a**–**e** in almost quantitative yields. The target 4-haloaryl-5-(thiophen-2-yl or 5-bromothiophene-2-yl)-2,4-dihydro-3*H*-1,2,4-triazole-3-thiones **5a**–**e** were obtained in good overall yields by cyclization of compounds **4a**–**e** via heating in 10% aqueous sodium hydroxide for two hours (Figure 1, Table 1).

The ^1^H NMR spectra of compounds **5a**–**e** showed the NH protons as broad singlets at *δ* 11.45–11.98 ppm and their ^13^C NMR spectra showed the C=S carbons at *δ* 168.02–169.79 ppm, confirming the existence of these compounds as the thione tautomers **A** rather than the thiol tautomers **B**. Full details of ^1^H NMR and ^13^C NMR spectral data of compounds **5a**-**e**, which were in full agreement with their structures, shown in Section 3.2.

5-Substituted-2,4-dihydro-3*H*-1,2,4-triazole-3-thiones were reported to react with primary or secondary amines and formaldehyde to yield the corresponding 2-aminomethyl-2,4-dihydro-3*H*-1,2,4-triazole-3-thiones (*N*-Mannich bases) [37,38,39,40,41,42,43]. Thus, compounds **5a** and **5b** were reacted with piperidine, morpholine, thiomorpholine or 1-substituted piperazines and 37% formaldehyde solution in ethanol to yield the corresponding *N*-Mannich bases **6a**–**e** and **7a**–**e** in good yields. The *N*-Mannich bases **8**, **9**, **10a** and **10b** were similarly prepared via the reaction of compound **5a** with 4-phenylpiperidine, 1,2,3,4-tetrahydroisoquinoline, *N*-methylaniline or *N*-benzylaniline and formaldehyde, respectively (Figure 2, Table 1).

The common ^1^H NMR spectral features of the *N*-Mannich bases **6a**–**e**, **7a**–**e**, **8**, **9**, **10a** and **10b** are characterized by the presence of methylene bridge protons (NCH_2_N) as sharp peaks at *δ* 5.02–5.95 ppm. In addition, the ^13^C NMR spectra showed the methylene bridge carbons at *δ* 66.52–72.70 ppm. Meanwhile, the cyclic thione carbons were shown at *δ* 169.26–170.64 ppm. Full details of ^1^H NMR and ^13^C NMR spectral data of compounds **6a**–**e**, **7a**–**d**, **8**, **9**, **10a** and **10b**, which were in full agreement with their structures, are shown in Section 3.3.

### 2.2. Single-Crystal XRD Study and Structural Insights

The crystallographic data and refinement parameters for compounds **5a** and **5b** are summarized in Appendix A. X-ray analysis revealed that both compounds crystallize in the triclinic crystal system with the space group P-1. The ORTEP representation of compounds **5a** and **5b** is depicted in Figure 3a and Figure 3b, respectively. In compound **5a**, the 1,2,4-triazole ring forms coplanarity with the mean plane of the thiophene ring (2.67°), while such a coplanarity is not observed in compound **5b** (12.39°). The dihedral angle between the mean planes of the triazole ring and the substituted phenyl ring is 73.97° and the corresponding angle is found to be 87.71° in compound **5b**. The structures of compounds **5a** and **5b** are overlaid with respect to the triazole ring, indicating a slight rotation around the thiophene and substituted phenyl ring (Figure 3c).

In the crystalline state, both compounds showed similar packing features. Molecules **5a** and **5b** are arranged in a ladder-like architecture, as shown in Figure 4. The ladder-like pattern is seen along the crystallographic bc plane in **5a**, whereas a similar pattern is observed along the crystallographic ac plane in **5b** (Figure 4a,b). The intermolecular interactions that stabilize the crystal structures of **5a** and **5b** are summarized in Table 2. The former structure stabilizes with intermolecular N–H···S, C–H···N and C–H···F hydrogen bonds and a C–S···S chalcogen bond. The latter structure also stabilizes with the above interactions in addition to the C12–H12···C5 hydrogen bond and hetero halogen bond (Br···F). This halogen is established due to the presence of an additional organic Br substituent in **5b**.

In compound **5a**, the amine group of the triazole ring is involved in N–H···S hydrogen bonds with the thione group producing $R_{2}^{2}$(8) motif. The adjacent N–H···S hydrogen-bonded dimers are interconnected by C–S···S chalcogen bonds, involving the thiophene S atom acting as a donor (σ-hole) and the thione S atom acting an acceptor. The N–H···S hydrogen bond and C–S···S chalcogen bond generate a supramolecular sheet, as shown in Figure 4c. The same kind of supramolecular sheet is also formed in **5b**, utilizing N–H···S hydrogen bond and C–S···S chalcogen bond. This supramolecular sheet is further supported by intermolecular C–H···C(π) interactions (involving thiophene and substituted phenyl rings) as shown in Figure 4d.

In addition to the above interactions, both compounds **5a** and **5b** also exhibit a dimer, which is formed by intermolecular C–H···N interaction with the graph set motif of $R_{2}^{2}$(12) (Figure 5a,b). In compound **5a**, the adjacent dimers (mediated by C–H···N interaction) are further interlinked by an intermolecular C–H···N interaction. The triazole N3 atom is involved in three-centered interactions (Figure 5a) and these C–H···N interactions generate alternate $R_{2}^{2}$(12)-$R_{4}^{2}$(10)-$R_{2}^{2}$(12) motifs. In compound **5b**, only $R_{2}^{2}$(12) motif is formed and it is not extended further (Figure 5b). It is of interest to note that the halogen bond is formed between the Br1 and F1 atoms. This interaction links the **5b** molecules into a chain, as shown in Figure 5c.

The crystallographic data and refinement parameters for compounds **6a**, **6d** and **10a** are presented in Appendix A. The X-ray analysis revealed that compounds **6a**, **6d** and **10a** crystallize in the monoclinic crystal system with the space group *P*2_1_/c. In the asymmetric unit of **6a**, there are two crystallographically independent molecules (molecules A and B), while one molecule is present in the asymmetric units of compounds **6d** and **10a**. In all three cases, the thiophene ring was disordered over two positions (~180° rotation).

In both the crystallographically independent molecules of compound **6a**, the thiophene ring was disordered in two orientations with a refined occupancy ratio of 0.613 (4):0.387 (4) in molecule A. The corresponding occupancy ratio of 0.579 (4):0.421 (4) in molecule B. The ORTEP diagram shows the major and minor disordered components of molecules A and B of **6a** (Figure 6).

The major disordered component was used for further analysis. The major disordered components of molecules A and B overlaid very well with the RMSD value of 0.07 Å. In molecule A, the thiophene ring is twisted with respect to the mean plane of the central triazole ring with a dihedral angle of 31.53° (29.52° in molecule B) and the observed twist is relatively large compared to compounds **5a** and **5b**. The dihedral angle between the mean planes of the triazole and the fluorophenyl rings is 70.24° (70.74° in molecule B). Furthermore, the piperidine ring exhibits a typical chair conformation. The mean plane formed by the piperidine ring and the central triazole ring makes an angle of 59.75° (62.58° in molecule B).

Figure 7a shows the columnar packing mode of compound **6a** along the crystallographic *bc* plane. The intermolecular interactions (C–H···S/F/π interactions and a short S_(lp)_···C_(π)_ contact) that stabilize the crystal structure of **6a** are summarized in Table 3. In the solid state, molecule A and its counterparts related to symmetry generate a supramolecular chain by intermolecular C–H···S, C–H···π (involving thiophene ring π-center as an acceptor) interactions and a short S_(lp)_···C_(π)_ contact (Figure 7b). A similar type of supramolecular chain and interactions formed between molecule B and its partners related to symmetry (Figure 7c). The intermolecular S_(lp)_···C_(π)_ contact was also observed earlier in 1,2,4-triazole derivatives [44]. Furthermore, molecule B interacts with its symmetry equivalent molecule through intermolecular C–H···S and C–H···F interactions to produce a molecular dimer (Figure 7d). Molecular A and molecule B interact via intermolecular C–H···F interactions, as shown in Figure 7e.

In compound **6d**, the thiophene ring was disordered in two orientations with a refined occupancy ratio of 0.667 (5):0.333 (5). The ORTEP diagram shows the major and minor disordered components of **6d** (Figure 8). The major disordered component was used for further analysis. The morpholine ring adopts a typical chair conformation. The thiophene and the central triazole ring are oriented at an angle of 17.92°. The disubstituted phenyl ring is inclined at an angle of 83.58° with respect to the mean plane of the triazole ring. The corresponding angle is 72.31° between the mean planes of the triazole and morpholine rings. As shown in Figure 8c, **6d** molecules were columnarly packed along the crystallographic *ac* plane with an interesting packing feature. In each column, the morpholine rings come closer to each other. The disubstituted phenyl rings are closer together between two adjacent columns. The intermolecular interactions that stabilize the crystal structure of **6d** are summarized in Table 3. In the solid state, centrosymmetrically related molecules form a molecular dimer stabilized by O_(lp)_···C_(π)_ contacts (involving morpholine ring and triazole rings) (Figure 8d). This structure also exhibits a halogen bond involving Br and F atoms that link the molecules into a supramolecular chain (Figure 8e). In addition to these interactions, the intermolecular C–H···S/F interactions generate a molecular dimer of **6d**, as shown in Figure 8f.

In compound **10a** also, the thiophene ring was disordered over two orientations with a refined occupancy ratio of 0.634 (2):0.366 (2). The ORTEP diagram shows the major and minor disordered components of **10d** (Figure 9). The major disordered component was used for further analysis. The dihedral angle between the mean planes of the thiophene and the central triazole rings is 27.17°. The triazole ring makes an angle of 79.50° and 64.86° with respect to the mean plane of the fluorophenyl and phenyl rings, respectively. Figure 9a shows the columnar packing mode of compound **10a** along the crystallographic *ac* plane. The intermolecular interactions that stabilize the crystal structure of **10a** are summarized in Table 3.

In the crystalline state of compound **10a**, molecules related by center of inversion (−x, −y, −z) form a dimer through intermolecular C–H···F interactions (involving the thiophene and the F-substituted rings). In addition, a molecular dimer is formed by intermolecular C–H···N interaction and a chalcogen bond between the S atom of the thiophene ring and one of the C atoms of the phenyl ring. A similar type (C–S···π) of chalcogen bond was observed in the 1,2,4-triazolo [3,4-*b*][1,3,4]thiadiazole derivative [45]. As shown in Figure 10a, this dimer and the former center of the inversion-related dimer are alternately linked, generating a supramolecular chain. Further, there are other two dimers also observed in **10a**, which help to stabilize the crystal structure. One of the dimers stabilizes with chalcogen bonds involving the thione S atom and centroid of the F-substituted phenyl ring (Figure 10b) and other dimer stabilizes with C–H···S/F interactions in which methyl and F-substituted phenyl ring act as a donor for these interactions (Figure 10c).

### 2.3. In Vitro Antimicrobial Activity

The in vitro antibacterial and antifungal activities of the newly synthesized compounds **5a**–**e**, **6a**–**e**, **7a**–**d**, **8**, **9**, **10a** and **10b** were evaluated against a panel of standard pathogenic bacterial and fungal strains of the Institute of Fermentation of Osaka (IFO), namely *Staphylococcus aureus* IFO 3060, *Bacillus subtilis* IFO 3007 and *Micrococcus luteus* IFO 3232 (Gram-positive bacteria); *Escherichia coli* IFO 3301 and *Pseudomonas aeruginosa* IFO 3448 (Gram-negative bacteria); and the pathogenic fungi *Candida albicans* IFO 0583, *Aspergillus oryzae* IFO 4177 and *Aspergillus niger* IFO 4414. The initial screening was carried out using the semi-quantitative agar disk-diffusion method using the Mueller–Hinton agar medium [46]. The results of the initial antimicrobial screening of compounds **5a**–**e**, **6a**–**e**, **7a**–**e**, **8**, **9**, **10a** and **10b** (200 μg/disc); the antibacterial antibiotics Ampicillin trihydrate and Ciprofloxacin; and the antifungal drug Fluconazole (100 μg/disc) are shown in Table 4.

The results showed variable grades of inhibition against the tested microorganisms. In general, marked antibacterial activity was shown by compounds **5a**, **5b**, **5e**, **5f**, **6a**–**e**, **7a**–**d**, **8**, **9**, **10a** and **10b**, which showed growth inhibition zones ≥ 18 mm, particularly against the tested Gram-positive bacteria *S. aureus* and *B. subtilis*. Meanwhile, marked inhibitory activity was displayed by compounds **5a**, **5b**, **5e**, **6a**, **6d**, **6e**, **6d**, **6e**, **7a**, **7d** and **8** against the Gram-negative bacteria *E. coli* with lower activity against *P. aeuroginosa*. All the tested compounds were found to be almost inactive against all the tested fungal strains.

In the 4-aryl-5-(thiophen-2-yl or 5-bromothiophen-2-yl)-2,4-dihydro-3*H*-1,2,4-triazole-3-thiones series **5a**–**e**, it was noticed that replacement of the 4-fluorophenyl, 2-bromo-4-fluorophenyl and 4-bromophenyl substituents at position 4 of the core triazole ring with 3- or 4-chlorophenyl moieties (compounds **5c** and **5d**) greatly deteriorated the antibacterial activity. In addition, an additional bromine atom on the thiophene ring (compounds **5c**–**e**) did not influence the potency and spectrum of the antibacterial activity.

The antibacterial activity of the 2-aminomethy derivatives **6a**–**e**, **7a**–**d**, **8**, **9**, **10a** and **10b** (*N*-Mannich bases) was generally superior to their precursors **5a** and **5b**. In the piperidinomethyl, morpholinomethyl and thiomorpholinomethyl derivatives **6a**–**e**, it was observed that replacement of the 4-fluorophenyl substituent at position 4 of the core triazole ring with 2-bromo-4-fluorophenyl enhanced activity against *M. luteus*. Potent and broad-spectrum antibacterial activity was attained in the piperzinomethyl derivatives **7a**–**d**. As noticed in compounds **6a**–**e**, replacement of the 4-fluorophenyl substituent at position 4 of the core triazole ring with the 2-bromo-4-fluorophenyl moiety in compounds **7a**–**d** enhanced activity against *M. luteus*. The optimum antibacterial activity was shown by the 4-methylpiperazino and the 4-(2-methoxyphenyl)piperazino analogs **7a** and **7d**, which exhibited potent antibacterial activity against *S. aureus*, *B. subtilis* and *E. coli* and retained moderate activity against *Pseudomonas aeuroginosa* (growth inhibition zones 14–17 mm). The antibacterial activity of the 4-phenylpiperazino derivative **8** and its annulated derivative **9** was almost similar, with potent activity against *S. aureus*, *B. subtilis* and *E. coli*. The *N*-methylanilino derivative **10a** displayed potent activity against the Gram-positive bacteria *S. aureus* and *B. subtilis*. Meanwhile, the activity of the *N*-benzylanilino analog **10b** was slightly altered against the Gram-positive bacteria with moderate potency (growth inhibition zones 10–13 mm) against Gram-negative bacteria *E. coli*.

The minimal inhibitory concentrations (MIC) and the minimal bactericidal concentrations (MBC) for the active compound against the same microorganism used in the primary screening were carried out using the microdilution susceptibility method in Mueller–Hinton Broth and Sabouraud Liquid Medium [47,48]. The MIC and MBC values of compounds **5a**, **5b**, **5e**, **6b**–**e**, **7a**–**d**, **8**, **9** and **10a** (which showed inhibition zones > 20 mm) and the antibacterial antibiotics Ampicillin trihydrate and Ciprofloxacin (Table 5) were consistent with the results of the initial screening.

Antimicrobial standards state that an agent is typically categorized as fungicidal or bactericidal if the MIC/MBC ratio is less than 4 [49]. The MIC/MBC ratio for all the active compounds **5a**, **5b**, **5e**, **6b**–**e**, **7a**–**d**, **8**, **9** and **10a** were found to be less than 4. Accordingly, these compounds are considered potential antibacterial candidates for further studies.

### 2.4. In Vitro Anti-Proliferative Activity

Compounds **5a**–**e**, **6a**–**e**, **7a**–**e**, **8**, **9**, **10a** and **10b** were tested for in vitro anti-proliferative activity against four human cancer cell lines, namely hepatocellular carcinoma (HePG-2), breast cancer (MCF-7), prostate cancer (PC-3) and colorectal cancer (HCT-116), by means of the 3-[4,5-dimethylthiazol-2-yl]-2,5-diphenyltetrazolium bromide (MTT) colorimetric assay [50,51]. Table 6 presents the results of the anti-proliferative activity of compounds **5a**–**e**, **6a**–**e**, **7a**–**e**, **8**, **9**, **10a** and **10b** as well as the anticancer drug Doxorubicin [52].

As indicated by the anti-proliferative activity results, the tested compounds displayed variable degrees of activity against the tested cancer cell lines. Generally speaking, the compounds exhibited potent anti-proliferative activity against HePG-2 and MCF-7. In addition, the *N*-Mannich bases **6a**–**e**, **7a**–**e**, **8**, **9**, **10a** and **10b** had higher activity than their precursors **5a** and **5b**. Compounds **5e**, **6a**–**e**, **7a**–**d** and **10a** exhibited potent activity with IC_50_ < 25 μM.

Within the 5-substituted-2,4-dihydro-3*H*-1,2,4-triazole-3-thiones **5a**–**e**, the optimal activity was attained by compound **5e**, which showed potent activity against HePG-2 and retained moderate activity (IC_50_ 26–50 μM) or weak activity (IC_50_ 51–100 μM) against MCF-7, PC-3 and HCT-116 cell lines.

In the 2-aminomethyl derivatives **6a**–**e**, **7a**–**d**, **8**, **9**, **10a** and **10b**, the anti-proliferative activity was mainly dependent on the nature of the haloaryl substituents at position 4 of the core triazole nucleus and the peripheral amino moieties. The anti-proliferative activity of the 4-(2-bromo-4-fluorophenyl) analogs (**6d**, **6e**, **7c** and **7d**) was higher than the 4-(4-fluorophenyl) analogs (**6a**–**c**, **7a** and **7b**). Regarding the peripheral amino moieties, the anti-proliferative activity of the piperidine, morpholine and piperazine derivatives (**6a**–**e** and **7a**–**d**) was generally higher than their 4-phenylpiperidine **8**, tetrahydroisoquinoline **9**, *N*-methylaniline **10a** and *N*-benzylaniline **10b** analogs. However, *N*-methylaniline **10a** retained good activity against HePG-2, moderate activity against MCF-7, weak activity against PC-3 and lacked activity against HCT-116 cell lines.

### 2.5. Molecular Docking Analysis

To corroborate the in vitro antibacterial activity of the most active compounds (**5e**, **6b**, **6d** and **7d**), we performed a molecular docking simulation to predict the favorable pose of these compounds at the active site of the DNA gyrase subunit B of *Staphylococcus aureus*. The CB-Dock2 program combines cavity detection and molecular docking with Autodock vina for the given protein target [53,54,55]. In this study, the 3D structure of DNA gyrase subunit B (PDB ID: 3G75) from *Staphylococcus aureus* was retrieved from the protein data bank. The co-crystallized ligand (ligand ID: B38) was used as a control to assess the binding affinity of select title compounds. Figure 11 shows the predicted pose of these compounds and overlaps with the position of the control inhibitor B38 (4-methyl-5-[3-(methylsulfanyl)-1*H*-pyrazol-5-yl]-2-thiophen-2-yl-1,3-thiazole). The docking score of these compounds and the co-crystallized inhibitor B38 is summarized in Table 7. The result suggests that compound **7d** showed a relatively better affinity for the target DNA gyrase subunit B compared to our other compounds and the control inhibitor. The intermolecular interactions between protein–ligand complexes were analyzed using the predicted poses with the PLIP web server [56].

Compound **5e** establishes four hydrophobic interactions and one hydrogen bond. Residues Asn 54, Ile 86 and Ile 175 are involved in hydrophobic interactions. The backbone O atom and the triazole N atom are engaged in a hydrogen bond. It is noted that in the remaining compounds (**6b**, **6d**, **7a** and **7d**), the active site residues Asn 54 are involved in a hydrogen bond with the ligands, mostly with the bulky substitution region introduced in the triazole nucleus. The residues Glu 50 and Val 131 are involved in hydrophobic contacts with **6b**. Similarly, residues Glu 58 and Pro 87 are hydrophobic, whereas the side chain O atom of Asp 57 acts as an acceptor for a halogen bond with the Br atom in **6d**. In **7a**, the side chain of Asp 81 (O atom) establishes a short contact with F of **7a**. Furthermore, Ile 86 and Asn 54 also participate in hydrophobic interactions. As seen in Figure 11f, there is a relatively greater number of interactions between **7d** and the active site residues; therefore, the affinity is slightly stronger compared to other compounds. In this complex, the methoxy group is involved in a hydrogen bond with the side chain of Asn 54. The other residues are involved in hydrophobic contacts.

We also explored the anti-proliferative potential of compounds **6d**, **6e** and **7d** against one of the cancer targets, namely cyclin-dependent kinase 2 (CDK2), a class of cell cycle regulators implicated in multiple cancers [57]. The 3D structure of human CDK2 was retrieved from the protein data bank with accession ID: 8OY2. This protein was complexed with an inhibitor molecule (ligand id: W5W; (1S,2S,11aS)-1-methoxy-1,4,7,10-tetramethyl-2,9-bis(oxidanyl)-2,11a-dihydrobenzo[*b*][1,4]benzodioxepine-3,6-dione. The molecular docking simulation correctly identified the active site and placed the ligand at the active site. The experimental conformation of W5W and the predicted pose of this molecule overlap very well, indicating the effectiveness of the program. The docking score revealed that compounds **6d**, **6e** and **7d** showed a relatively better binding affinity than the control inhibitor W5W (Table 8). The predicted pose of these molecules was used to analyze the intermolecular interactions formed between active site residues and the ligand molecules (Figure 12).

Compound **6d** interacts with the active site residues via hydrophobic (Asn 132 and Val 18) interactions and a salt bridge between the N atom of the morpholine and carboxylate of the Asp 86. The same set of residues is also involved in interactions with compound **6e**. Relatively, compound **7d** establishes a greater number of contacts with the active site residues. Residues Leu 134, Ala 31, Ile 10, Val 18, Ala 149 and Asp 127 are involved in hydrophobic interactions, while Asp 145 participates in salt bridge interaction with one of the N atoms of the piperazine.

Taken as a whole, compound **7d** exhibits both antibacterial and anti-proliferative potentials, as revealed by in vitro and in silico studies.

## 3. Materials and Methods

### 3.1. General Information

Melting points (°C, uncorrected) were determined in open-glass capillaries using a Stuart SMP30 electro–thermal melting point apparatus (Nottingham, UK). Nuclear magnetic resonance (NMR) spectra were determined in CDCl_3_ on Bruker RMN AV600 and Bruker Avance III HD FT-high resolution NMR instruments (Billerica, MA, USA) at *δ* 600.15 MHz for ^1^H and 150.36 MHz for ^13^C, *δ* 400.20 MHz for ^1^H and 100.64 MHz for ^13^C, respectively. Elemental analyses (C, H, N and S) were in agreement with the proposed structures within ±0.3% of the theoretical values (Appendix A). Monitoring of the reactions and checking of the purity of the final products was carried out with thin layer chromatography (TLC) using silica gel-precoated aluminum sheets 60 F_254_ (Merck, Darmstadt, Germany) and visualization with ultraviolet light (UV) at 365 and 254 nm. All chemicals and solvents were purchased from commercial suppliers and used without additional purification. The reference drugs Ampicillin trihydrate (CAS # 7177-48-2) and Ciprofloxacin (CAS # 85721-33-1), Fluconazole (CAS # 86386-73-4) and Doxorubicin (CAS 23214-92-8) were purchased from Sigma-Aldrich Chemie GmbH (Taufkirchen, Germany). The synthesis of compound **5a** was previously reported via the reaction of ethyl thiophene-2-carboxylate with thiosemicarbazide in anhydrous methanol in the presence of sodium methylate followed by heating for 5 min [58].

### 3.2. General Procedure for the Synthesis of 4-Haloaryl-5-(thiophen-2-yl or 5-Bromothiophen-2-yl)-2,4-dihydro-3H-1,2,4-triazole-3-thiones **5a**–**e**

The appropriate haloaryl isothiocyanate (0.01 mole) was added to a solution of thiophene-2-carbohydrazide **3a** or 5-bromothiophene-2-carbohydrazide **3b** (0.01 mole) in ethanol (15 mL) and the mixture was heated under reflux with stirring for 1 h. The solvent was distilled off in vacuo to yield the intermediates *N*-aryl-2-(thiophene-2-carbonyl or 5-bromothiophene-2-carbonyl)hydrazine-1-carbothioamides **4a**–**e** in almost quantitative yields. Aqueous sodium hydroxide solution (10%, 15 mL) was added to compounds **4a**–**e** and the mixture was heated under reflux for 2 h and then filtered hot. On cooling, the mixture was acidified with hydrochloric acid and the precipitated crude product was filtered, washed with water, dried and crystallized from ethanol to yield the target products **5a**–**e**.

4-(4-Fluorophenyl)-5-(thiophen-2-yl)-2,4-dihydro-3*H*-1,2,4-triazole-3-thione **5a**. ^1^H NMR (600.15 MHz): *δ* 6.86 (d, 1H, Thiophene-H, *J* = 3.9 Hz), 6.94 (t, 1H, Thiophene-H, *J* = 3.9 Hz), 7.26–7.40 (m, 5H, Ar-H and Thiophene-H), 11.45 (br. s, 1H, NH). ^13^C NMR (150.91 MHz): *δ* 117.22, 126.38, 127.66, 129.12, 129.84, 130.77, 144.41, 147.05, 162.71 (Ar-C, Thiophene-C and Triazole-C5), 169.78 (C=S).

4-(2-Bromo-4-fluorophenyl)-5-(thiophen-2-yl)-2,4-dihydro-3*H*-1,2,4-triazole-3-thione **5b**. ^1^H NMR (400.20 MHz): *δ* 6.90 (d, 1H, Thiophene-H, *J* = 4.0 Hz), 6.96 (t, 1H, Thiophene-H, *J* = 4.0 Hz), 7.28–7.44 (m, 4H, Ar-H and Thiophene-H), 11.65 (br. s, 1H, NH). ^13^C NMR (100.64 MHz): *δ* 116.94, 122.03, 124.68, 128.05, 129.78, 132.58, 135.38, 143.95, 149.16, 162.37, 164.36 (Ar-C, Thiophene-C and Triazole-C5), 169.79 (C=S).

5-(5-Bromothiophen-2-yl)-4-(3-chlorophenyl)-2,4-dihydro-3*H*-1,2,4-triazole-3-thione **5c**. ^1^H NMR (400.20 MHz): *δ* 7.08 (d, 1H, Thiophene-H, *J* = 4.0 Hz), 7.10–7.22 (m, 4H, Ar-H and Thiophene-H), 7.38 (s, 1H, Ar-H), 11.68 (br. s, 1H, NH). ^13^C NMR (100.64 MHz): *δ* 116.20, 126.0, 127.22, 128.06, 129.36, 130.12, 130.84, 131.88, 133.0, 135.22, 138.42 (Ar-C, Thiophene-C and Triazole-C5), 168.62 (C=S).

5-(5-Bromothiophen-2-yl)-4-(4-chlorophenyl)-2,4-dihydro-3*H*-1,2,4-triazole-3-thione **5d**. ^1^H NMR (400.20 MHz): *δ* 7.01 (d, 1H, Thiophene-H, *J* = 3.9 Hz), 7.08–7.36 (m, 5H, Ar-H and Thiophene-H), 11.80 (br. s, 1H, NH). ^13^C NMR (100.64 MHz): *δ* 115.98, 126.68, 127.0, 128.02, 129.24, 133.10, 136.66, 139.58 (Ar-C, Thiophene-C and Triazole-C5), 168.44 (C=S).

4-(4-Bromophenyl)-5-(5-bromothiophen-2-yl)-2,4-dihydro-3*H*-1,2,4-triazole-3-thione **5e**. ^1^H NMR (400.20 MHz): *δ* 7.33–7.42 (m, 2H, Ar-H and Thiophene-H), 7.04 (d, 1H, Thiophene-H, *J* = 4.0 Hz), 7.32 (d, 2H, Ar-H), 7.35–7.37 (m, 3H, Ar-H and Thiophene-H), 11.98 (br. s, 1H, NH). ^13^C NMR (100.63 MHz): *δ* 114.98, 123.68, 127.0, 128.02, 129.24, 133.10, 135.12, 136.66, 139.58 (Ar-C, Thiophene-C and Triazole-C5), 168.02 (C=S).

### 3.3. General Procedure for the Synthesis of 2-Aminomethyl-4-aryl-5-(thiophen-2-yl or 5-Bromothiophen-2-yl)-2,4-dihydro-3H-1,2,4-triazole-3-thiones **6a**–**e**, **7a**–**d**, **8**, **9**, **10a** and **10b**

The appropriate secondary amine piperidine, morpholine, thiomorpholine, 1-methylpiperazine, 1-phenylpiperazine, 1-(2-methoxyphenyl)lpiperazine, 4-phenylpiperidine, 1,2,3,4-tetrahydroisoquinoline, *N*-methylaniline or *N*-benzylaniline (0.01 mole) and 37% formaldehyde solution (1.0 mL) were added to a hot solution of 4-(4-fluorophenyl)-5-(thiophen-2-yl)-2,4-dihydro-3*H*-1,2,4-triazole-3-thione **5a** or 4-(2-bromo-4-fluorophenyl)-5-(thiophen-2-yl)-2,4-dihydro-3*H*-1,2,4-triazole-3-thione **5b** (0.01 mole) in ethanol (15 mL) and the mixture was heated under reflux for 10 min then stirred at room temperature for 5 h and allowed to stand overnight. The precipitated crude products were filtered, washed with water, dried and crystallized from ethanol or aqueous ethanol.

4-(4-Fluorophenyl)-2-(piperidin-1-ylmethyl)-5-(thiophen-2-yl)-2,4-dihydro-3*H*-1,2,4-triazole-3-thione 6a. ^1^H NMR (600.15 MHz): *δ* 1.41–1.44 (m, 2H, Piperidine-CH_2_), 1.59–1.63 (m, 4H, Piperidine-CH_2_), 2.85 (t, 4H, Piperidine-CH_2_, *J* = 5.4 Hz), 5.88 (s, 2H, NCH_2_N), 6.88 (d, 1H, Thiophene-H, *J* = 2.0 Hz), 6.93 (t, 1H, Thiophene-H, *J* = 2.0 Hz), 7.25–7.27 (m, 2H, Ar-H), 7.36–7.38 (m, 3H, Ar-H and Thiophene-H). ^13^C NMR (150.36 MHz): *δ* 23.80, 25.99, 51.87 (Piperidine-C), 71.04 (NCH_2_N), 117.09, 126.71, 127.58, 129.0, 130.78, 130.89, 130.94, 144.82, 162.59, 164.26 (Ar-C, Thiophene-C and Triazole-C5), 170.38 (C=S).

4-(4-Fluorophenyl)-2-(morpholinomethyl)-5-(thiophen-2-yl)-2,4-dihydro-3*H*-1,2,4-triazole-3-thione 6b. ^1^H NMR (400.20 MHz): *δ* 2.68 (t, 4H, Morpholine-CH_2_, *J* = 5.0 Hz), 3.18 (t, 4H, Morpholine-CH_2_, *J* = 5.0 Hz), 5.20 (s, 2H, NCH_2_N), 6.90–6.98 (m, 2H, Ar-H and Thiophene-H), 7.16–7.34 (m, 2H, Ar-H and Thiophene-H), 7.39–7.39 (m, 3H, Ar-H and Thiophene-H). ^13^C NMR (100.46 MHz): *δ* 50.70, 71.19 (Morpholine-C), 71.0 (NCH_2_N), 116.42, 121.52, 127.73, 128.60, 129.22, 132.20, 132.84, 144.58, 162.14, 164.20 (Ar-C, Thiophene-C and Triazole-C5), 169.64 (C=S).

4-(4-Fluorophenyl)-2-(thiomorpholinomethyl)-5-(thiophen-2-yl)-2,4-dihydro-3*H*-1,2,4-triazole-3-thione 6c. ^1^H NMR (400.20 MHz): *δ* 2.70 (t, 4H, Thiomorpholine-CH_2_, *J* = 4.0 Hz), 3.18 (t, 4H, Morpholine-CH_2_, *J* = 4.0 Hz), 5.21 (s, 2H, NCH_2_N), 6.84 (d, 1H, Thiophene-H, *J* = 4.0 Hz), 6.92 (t, 1H, Thiophene-H, *J* = 4.0 Hz), 7.24–7.38 (m, 5H, Ar-H and Thiophene-H). ^13^C NMR (100.64 MHz): *δ* 28.05, 52.84 (Thiomorpholine-C), 71.39 (NCH_2_N), 117.22, 126.48, 127.73, 129.20, 130.87, 130.96, 145.10, 162.24. 164.74 (Ar-C, Thiophene-C and Triazole-C5), 170.32 (C=S).

4-(2-Bromo-4-fluorophenyl)-2-(morpholinomethyl)-5-(thiophen-2-yl)-2,4-dihydro-3*H*-1,2,4-triazole-3-thione 6d. ^1^H NMR (600.15 MHz): *δ* 2.87–97 (m, 4H, Morpholine-CH_2_), 3.72 (t, 4H, Morpholine-CH_2_, *J* = 4.0 Hz), 5.34 (d, 2H, NCH_2_N, *J* = 6.0 Hz), 6.94–6.98 (m, 2H, Ar-H and Thiophene-H), 7.27–7.30 (m, 2H, Ar-H and Thiophene-H), 7.39–7.55 (m, 2H, Ar-H and Thiophene-H). ^13^C NMR (150.36 MHz): *δ* 50.67, 66.93 (Morpholine-C), 69.80 (NCH_2_N), 116.47, 121.57, 124.74, 126.19, 127.77, 128.66, 129.26, 130.29, 144.63, 162.37, 164.07 (Ar-C, Thiophene-C and Triazole-C5), 169.98 (C=S).

4-(2-Bromo-4-fluorophenyl)-2-(thiomorpholinomethyl)-5-(thiophen-2-yl)-2,4-dihydro-3*H*-1,2,4-triazole-3-thione **6e**. ^1^H NMR (400.20 Hz): *δ* 2.64 (t, 4H, Thiomorpholine-CH_2_, *J* = 4.7 Hz), 3.11–3.21 (m, 4H, Thiomorpholine-CH_2_), 5.26 (d, 2H, NCH_2_N, *J* = 6.2 Hz), 6.90–6.94 (m, 2H, Ar-H and Thiophene-H), 7.22–7.26 (m, 1H, Ar-H), 7.35 (d, 1H, Thiophene-H, *J* = 4.5 Hz), 7.36–51 (m, 2H, Ar-H). ^13^C NMR (100.64 Hz): *δ* 28.75, 52.86 (Thiophene-C), 71.35 (NCH_2_N), 116.86, 121.68, 124.85, 127.90, 128.76, 129.39, 132.37, 132.45, 144.74, 162.31, 164.35 (Ar-C, Thiophene-C and Triazole-C5), 169.80 (C=S).

4-(4-Fluorophenyl)-2-[(4-methylpiperazin-1-yl)methyl]-5-(thiophen-2-yl)-2,4-dihydro-3*H*-1,2,4-triazole-3-thione **7a**. ^1^H NMR (400.20 MHz): *δ* 1.76–1.91 (m, 7H, CH_3_ and Piperazine-CH_2_), 2.44–2.67 (m, 4H, Piperazine-CH_2_), 5.30 (s, 2H, NCH_2_N), 6.89 (d, 1H, Thiophene-H, *J* = 4.0 Hz), 6.95 (t, 1H, Thiophene-H, *J* = 4.0), 7.20–7.43 (m, 5H, Ar-H and Thiophene-H). ^13^C NMR (100.64 MHz): *δ* 33.53, 51.75 (Piperazine-C), 42.09 (CH_3_), 70.66 (NCH_2_N), 117.25, 126.30, 127.77, 128.56, 129.20, 130.97, 146.32, 162.30, 164.81 (Ar-C, Thiophene-C and Triazole-C5), 170.51 (C=S).

4-(4-Fluorophenyl)-2-[(4-phenylpiperazin-1-yl)methyl]-5-(thiophen-2-yl)-2,4-dihydro-3*H*-1,2,4-triazole-3-thione 7b. ^1^H NMR (400.20 Hz): *δ* 3.12 (s, 4H, Piperazine-CH_2_), 3.28 (t, 4H, Piperazine-CH_2_, *J* = 5.2 Hz), 5.30 (s, 2H, NCH_2_N), 6.84 (d, 1H, Thiophene-H, *J* = 3.6 Hz), 6.87–6.93 (m, 2H, Ar-H), 6.96 (d, 1H, Thiophene-H, *J* = 4.0 Hz), 7.23–7.26 (m, 5H, Ar-H and Thiophene-H), 7.33–7.37 (m, 3H, Ar-H). ^13^C NMR (100.63 Hz): *δ* 49.68, 50.55 (Piperazine-C), 69.82 (NCH_2_N), 116.67, 117.30, 126.52, 127.78, 129.30, 129.34, 130.77, 130.80, 130.94, 131.03, 145.18, 162.32, 164.83 (Ar-C, Thiophene-C and Triazole-C5), 170.64 (C=S).

4-(2-Bromo-4-fluorophenyl)-2-[(4-phenylpiperazin-1-yl)methyl]-5-(thiophen-2-yl)-2,4-dihydro-3*H*-1,2,4-triazole-3-thione **7c**. ^1^H NMR (400.20 Hz): *δ* 2.36–2.42 (m, 4H, Piperazine-CH_2_), 3.28–3.36 (m, 4H, Piperazine-CH_2_), 5.20 (d, 2H, NCH_2_N, *J* = 6.2 Hz), 6.90 (d, 1H, Thiophene-H, *J* = 3.8 Hz), 6.98–7.44 (m, 6H, Ar-H andThiophene-H), 7.47–7.67 (m, 4H, Ar-H and Thiophene-H). ^13^C NMR (100.63 Hz): *δ* 50.08, 52.40 (Piperazine-C), 72.70 (NCH_2_N), 114.02, 116.46, 120.68, 122.0, 124.48, 126.98, 127.50, 128.60, 130.98, 137.46, 145.22, 148.0, 162.90, 164.62 (Ar-C, Thiophene-C and Triazole-C5), 169.88 (C=S).

4-(2-Bromo-4-fluorophenyl)-2-{[(4-(2-methoxyphenyl)piperazin-1-yl]methyl}-5-(thiophen-2-yl)-2,4-dihydro-3*H*-1,2,4-triazole-3-thione **7d**. ^1^H NMR (400.20 MHz): *δ* 2.86 (t, 4H, Piperazine-CH_2_, *J* = 4.8 Hz), 3.20 (t, 4H, Piperazine-CH_2_, *J* = 4.8 Hz), 3.96 (s, 3H, OCH_3_), 5.02 (d, 2H, NCH_2_N, *J* = 6.0 Hz), 6.89–7.13 (m, 5H, Ar-H and Thiophene-H), 7.23–7.39 (m, 3H, Ar-H and Thiophene-H), 7.40–7.55 (m, 2H, Ar-H and Thiophene-H). ^13^C NMR (100.63 MHz): *δ* 51.86, 52.44 (Piperazine-C), 54.20 (OCH_3_), 70.28 (NCH_2_N), 114.04, 115.84, 121.10, 121.86, 123.0, 124.02, 124.98, 126.42, 128.46, 129.86, 130.46, 136.82, 139.90, 140.96, 148.20, 161.46, 163.02 (Ar-C, Thiophene-C and Triazole-C5), 169.40 (C=S).

4-(4-Fluorophenyl)-2-[(4-phenylpiperidin-1-yl)methyl]-5-(thiophen-2-yl)-2,4-dihydro-3*H*-1,2,4-triazole-3-thione **8**. ^1^H NMR (400.20 MHz): *δ* 1.76–1.91 (m, 5H, Piperidine-H), 2.67–2.74 (m, 2H, Piperidine-H), 3.36–3.41 (m, 2H, Piperidine-H), 5.30 (s, 2H, NCH_2_N), 6.89 (d, 1H, Thiophene-H, *J* = 4.0 Hz), 6.95 (t, 1H, Thiophene-H, *J* = 4.0 Hz), 7.20–7.40 (m, 10H, Ar-H and Thiophene-H). ^13^C NMR (100.64 MHz): *δ* 33.53, 42.10, 51.76 (Piperidine-C), 70.66 (NCH_2_N), 117.25, 126.30, 126.94, 127.77, 128.56, 129.20, 129.24, 130.97, 131.06, 145.07, 146.32, 162.30, 164.81 (Ar-C, Thiophene-C and Triazole-C5), 170.51 (C=S).

2-[(3,4-Dihydroisoquinolin-2(1*H*)-yl)methyl]-4-(4-fluorophenyl)-5-(thiophen-2-yl)-2,4-dihydro-3*H*-1,2,4-triazole-3-thione 9. ^1^H NMR (400.20 Hz): *δ* 2.96 (t, 2H, Isoquinoline-CH_2_, *J* = 6.0 Hz), 3.27 (t, 2H, Isoquinoline-CH_2_, *J* = 6.0 Hz), 4.16 (s, 2H, Isoquinoline-CH_2_), 5.45 (s, 2H, NCH_2_N), 6.89 (d, 1H, Thiophene-H, *J* = 3.2 Hz), 6.95 (t, 1H, Thiophene-H, *J* = 3.6 Hz), 7.13–7.15 (m, 5H, Ar-H), 7.26–7.40 (m, 5H, Ar-H and Thiophene-H). ^13^C NMR (100.63 Hz): *δ* 29.35, 48.72, 52.56 (Isoquinoline-CH_2_), 69.89 (NCH_2_N), 117.15, 125.72, 126.16, 126.54, 126.71, 127.68, 128.85, 129.22, 130.74, 130.90, 133.91, 134.48, 145.11, 162.22, 164.72 (Ar-C, Thiophene-C, Triazole-C5 and Triazole-C5), 170.49 (C=S).

4-(4-Fluorophenyl)-2-[(*N*-methylanilino)methyl]-5-(thiophen-2-yl)-2,4-dihydro-3*H*-1,2,4-triazole-3-thione **10a**. ^1^H NMR (600.15 MHz): *δ* 3.40 (s, 3H, CH_3_), 5.85 (s, 2H, NCH_2_N), 6.83–6.86 (m, 3H, Ar-H and Thiophene-H), 6.91 (t, 1H, Thiophene-H, *J* = 4.3), 7.18 (d, 2H, Ar-H, *J* = 8.4 Hz), 7.24–7.36 (m, 6H, Ar-H and Thiophene-H). ^13^C NMR (150.36 MHz): *δ* 39.56 (CH_3_), 66.52 (NCH_2_N), 113.60, 117.12, 117.31, 118.58, 126.58, 127.53, 129.11, 129.24, 130.88, 130.94, 145.34, 147.35, 162.63, 164.40 (Ar-C, Thiophene-C and Triazole-C5), 169.26 (C=S).

4-(4-Fluorophenyl)-2-[(*N*-benzylanilino)methyl]-5-(thiophen-2-yl)-2,4-dihydro-3*H*-1,2,4-triazole-3-thione 10b. ^1^H NMR (400.20 MHz): *δ* 5.10 (s, 2H, Benzylic CH_2_), 5.95 (s, 2H, NCH_2_N), 6.81–6.93 (m, 3H, Ar-H and Thiophene-H), 7.17–7.36 (m, 15H, Ar-H and Thiophene-H). ^13^C NMR (100.64 MHz): *δ* 55.13 (CH_2_), 64.97 (NCH_2_N), 114.05, 117.16, 118.87, 126.61, 126.99, 127.67, 129.26, 129.29, 129.52, 130.53, 130.94, 131.03, 138.54, 145.46, 147.02, 162.28, 164.78 (Ar-C, Thiophene-C and Triazole-C5), 169.39 (C=S).

### 3.4. Single-Crystal XRD Studies

Suitable single crystals for X-ray diffraction were obtained by slow evaporation of a solution of compounds **5a**, **5b**, **6a**, **6d** and **10a** in ethanol:chloroform (1:1, *v*/*v*) at room temperature. The crystal data and structure refinement parameters are shown in Appendix A (compounds **5a** and **5b**) and Appendix A (compounds **6a**, **6d** and **10a**). The X-ray intensity data were collected on a Rigaku OD SuperNova/Atlas area-detector diffractometer using Cu Kα radiation (λ = 1.54184 Å) from a micro-focus X-ray source (for compound **5b**) and on Excalibur, Ruby and Gemini diffractometers for the remaining crystals. Using *Olex2* [59], the structure was solved with the SHELXT small molecule structure solution program [60] and refined with the *SHELXL2018/3* program package [61] by full-matrix least-squares minimization on F^2^. In **5a** and **5b**, the amino H atom was located from a difference Fourier map and refined freely with its isotropic displacement parameters. In the remaining compounds, the H atoms were placed in calculated positions and were constrained to ride on their parent atoms, with *U*_iso_(H) = 1.2*U*_eq_(C). The methyl H atoms were constrained to an ideal geometry with *U*_iso_(H) = 1.5*U*_eq_(C) but were allowed to rotate freely about the C–C bonds. In compounds **6a**, **6d** and **10d**, the thiophene ring was disordered over two orientations. These structures were refined with suitable disorder models using the appropriate restraints and satisfactory models were obtained for these compounds. The crystal structure of **5b** was refined as a two-component twin. The twin matrix is (−1.000 0 0 0 −1 0 0.4610.838 1) and the twin scales are 0.759(3) and 0.241(3).

## Data Availability

The full crystallographic data could be obtained free of charge from the Cambridge Crystallographic Data Centre (www.ccdc.cam.ac.uk/data_request/cif, accessed on 30 January 2022) using the accession numbers, CCDC-2343567 (compound **5a**), CCDC-2343568 (compound **5b**), CCDC-2343570 (compound **6a**), CCDC-2343571 (compound **6d**) and CCDC-2343572 (compound **10a**).

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
