# Peer review of "Thiophene-Linked 1,2,4-Triazoles: Synthesis, Structural Insights and Antimicrobial and Chemotherapeutic Profiles†"

_pharmaceuticals, 2024, doi:10.3390/ph17091123_

Round 1

Reviewer 1 Report

Comments and Suggestions for Authors

The authors reported the synthesis of thiophene-Linked 1,2,4-Triazoles derivatives and their anti-bacterial, anti-fungal, and anti-cancer activities. Additionally, molecular docking studies were provided to give more information on some target proteins such as DNA gyrase and CDK2. Overall, the work is well-written and presented clearly. 

Just a minor point.

The title should have a colon instead of a period before the word "Synthesis" and it should have some keyword of "anti-microbial" because the authors tested for both microbial and cancer activities.

Thus, it should be Thiophene-Linked 1,2,4-Triazoles: Synthesis, Structural Insights, Anti-microbial and Chemotherapeutic Profiles 

In my opinion, it can be accepted after a minor revision. 

Author Response

Response to Referee

Referee 1

Quality of English Language

(  ) I am not qualified to assess the quality of English in this paper.

(  ) The English is very difficult to understand/incomprehensible.
(  ) Extensive editing of English language required.
(  ) Moderate editing of English language required.
(  ) Minor editing of English language required.

(X) English language fine. No issues detected.

  • Does the introduction provide sufficient background and include all relevant references? Yes
  • Is the research design appropriate? Can be improved
  • Are the methods adequately described? Yes
  • Are the results clearly presented? Can be improved
  • Are the conclusions supported by the results? Yes

Many thanks for your comments and very constructive suggestions.

Comments and Suggestions for Authors:

The authors reported the synthesis of thiophene-Linked 1,2,4-Triazoles derivatives and their anti-bacterial, anti-fungal, and anti-cancer activities. Additionally, molecular docking studies were provided to give more information on some target proteins such as DNA gyrase and CDK2. Overall, the work is well-written and presented clearly.

Just a minor point.

The title should have a colon instead of a period before the word "Synthesis" and it should have some keyword of "anti-microbial" because the authors tested for both microbial and cancer activities.

Thus, it should be Thiophene-Linked 1,2,4-Triazoles: Synthesis, Structural Insights, Anti-microbial and Chemotherapeutic Profiles 

In my opinion, it can be accepted after a minor revision.

Author's Response: The title of the manuscript was revised as suggested.

__________________________________________________________________________

Reviewer 2 Report

Comments and Suggestions for Authors

The synthesis of 2-aminomethyl-4- 23 haloaryl-2,4-dihydro-3H-1,2,4-triazole-3-thiones derivatives was performed obtaining 15 final compounds that have been evaluated on different biological systems, the authors report the results of some of these compounds, in addition to a crystallographic study, NMR analysis, and docking study are reported. In general, the manuscript has been presented in a clear way, however before being approved for publication it is recommended:

a.      Authors should review the manuscript in detail, as there are spelling errors.

b.      Line 60, is OSI-930 or OSI-980?

c.      Table 4, is NT or ND?

d.      Line 331

e.      The paragraph between lines 333 and 344 should be revised, there are repeated phrases

f.       Line 401, check

g.      Line 464, 6b bold

h.      The data reported for the 1H NMR in compound 5b, 6b, 6e, 7b, 7e are not correct, a number of atoms are being reported that do not correspond to the structure

i.       Check references 25 and 49

Author Response

Response to Referee

Referee 2

Quality of English Language

(X) I am not qualified to assess the quality of English in this paper.

(  ) The English is very difficult to understand/incomprehensible.
(  ) Extensive editing of English language required.
(  ) Moderate editing of English language required.
(  ) Minor editing of English language required.

(  ) English language fine. No issues detected.

  • Does the introduction provide sufficient background and include all relevant references? Yes
  • Is the research design appropriate? Yes
  • Are the methods adequately described? Yes
  • Are the results clearly presented? Can be improved
  • Are the conclusions supported by the results? Can be improved

Many thanks for your comments and very constructive suggestions.

Comments and Suggestions for Authors:

The synthesis of 2-aminomethyl-4- 23 haloaryl-2,4-dihydro-3H-1,2,4-triazole-3-thiones derivatives was performed obtaining 15 final compounds that have been evaluated on different biological systems, the authors report the results of some of these compounds, in addition to a crystallographic study, NMR analysis, and docking study are reported. In general, the manuscript has been presented in a clear way, however before being approved for publication it is recommended:

  1. Authors should review the manuscript in detail, as there are spelling errors.

Author's Response: The language and spelling errors were corrected in the revised version of the manuscript.

  1. Line 60, is OSI-930 or OSI-980?

Author's Response: The correct compound code is OSI-930 (as stated in the manuscript text). The compound code was corrected in Figure 2.

  1. Table 4, is NT or ND?

Author's Response: The abbreviation NT (Not Tested) is more suitable in Table 4 because the standard antibacterial antibiotics (Ampicillin trihydrate and Ciprofloxacin) were not tested against the pathogenic fungi, and the standard antifungal agent Fluconazole was not tested against the pathogenic bacteria. Meanwhile, the abbreviation ND (Not determined) in Table 5 is more suitable because as it includes the determination of the MIC and MBC of the highly active compounds.

  1. Line 331

Author's Response: See, the above response.

  1. The paragraph between lines 333 and 344 should be revised, there are repeated phrases.

Author's Response: The text was revised and clarified by addition of “in compounds 7a-d’.

  1. Line 401, check

Author's Response: The typing mistake was corrected “weak”.

  1. Line 464, 6b bold

Author's Response: Corrected as advised.

  1. The data reported for the 1H NMR in compound 5b, 6b, 6e, 7b, 7e are not correct, a number of atoms are being reported that do not correspond to the structure.

Author's Response: All the NMR data was thoroughly examined and revised. In the 13C NMR data, it should be noted that more than one carbon may be detected presented as one peak (2 carbons are magnetically equivalent). There is no compound No. 7e. Anyway, the 1H NMR and 13C NMR spectra were included in the supplementary file of the manuscript.

  1. Check references 25 and 49

Author's Response: The reference 25 is correct, the error in the compound code was corrected in page 2, line 59. The reference 49 was revised and the year “2006” was written in bold.

__________________________________________________________________________

Reviewer 3 Report

Comments and Suggestions for Authors

The article realised Nada A. El-Emam and col. is an interesting and well-conducted research that captures the reader's interest. However, I find that it still needs to be supplemented with some data to meet the necessary criteria for publication, as follows:

  • Correction of the reaction scheme in Figures 1, 2, and Table 1, because Y cannot be used in both the structure of compounds 5c-e and 6a-e, as it does not refer to the same substituent (Table 4 also needs to be corrected accordingly).
  • Completion of the IR absorption spectra, without the spectral characterization is not completed.

·         The authors need to specify which compounds are original to emphasize the originality of the research. For example, compounds 5a and 5b are known (https://pubchem.ncbi.nlm.nih.gov/) as

4-(4-fluorophenyl)-3-thiophen-2-yl-1H-1,2,4-triazole-5-thione and

4-(2-bromo-4-fluorophenyl)-3-thiophen-2-yl-1H-1,2,4-triazole-5-thione

Can the authors explain why they did not retain these IUPAC chemical names?

  • All NMR spectra should be presented in the Supplementary Materials.

 Minor observations: As the authors have written "in silico" with Latin characters, "in vitro" should also be written using the same characters.

Author Response

Response to Referee

Referee 3

Quality of English Language

(X) I am not qualified to assess the quality of English in this paper.

(  ) The English is very difficult to understand/incomprehensible.
(  ) Extensive editing of English language required.
(  ) Moderate editing of English language required.
(  ) Minor editing of English language required.

(  ) English language fine. No issues detected.

  • Does the introduction provide sufficient background and include all relevant references? Yes
  • Is the research design appropriate? Must be improved
  • Are the methods adequately described? Can be improved
  • Are the results clearly presented? Can be improved
  • Are the conclusions supported by the results? Can be improved

Many thanks for your comments and very constructive suggestions.

Comments and Suggestions for Authors:

The article realised Nada A. El-Emam and col. is an interesting and well-conducted research that captures the reader's interest. However, I find that it still needs to be supplemented with some data to meet the necessary criteria for publication, as follows:

  • Correction of the reaction scheme in Figures 1, 2, and Table 1, because Y cannot be used in both the structure of compounds 5c-e and 6a-e, as it does not refer to the same substituent (Table 4 also needs to be corrected accordingly).

Author's Response: The general structures of the compounds 5a-e, 6a-e, 7a-d and 10a,b which contain variables (X, Y and R) were added at the top of table 1 to facilitate the identification of each compound.

  • Completion of the IR absorption spectra, without the spectral characterization is not completed.

Author's Response: The described compounds are fully characterized by 1H NMR, 13C NMR, elemental analysis, in addition to single crystal X diffraction of 5 representative derivatives. The IR spectra are of limited value compared to the adopted methods of characterization.

  • The authors need to specify which compounds are original to emphasize the originality of the research. For example, compounds 5a and 5b are known (https://pubchem.ncbi.nlm.nih.gov/) as 4-(4-fluorophenyl)-3-thiophen-2-yl-1H-1,2,4-triazole-5-thione and 4-(2-bromo-4-fluorophenyl)-3-thiophen-2-yl-1H-1,2,4-triazole-5-thione. Can the authors explain why they did not retain these IUPAC chemical names?

Author's Response: We agree with the referee that the synthesis of compound 5a was previously reported, the reported synthesis is quite different from the method described in this manuscript. However, we referred to the previously reported method (Reference # 58). Compound 5b was not previously reported, the search results on the PubChem (https://pubchem.ncbi.nlm.nih.gov/) gave only some theoretical calculations on the structure without any literature about the chemical synthesis and characterization (see: https://pubchem.ncbi.nlm.nih.gov/compound/29074137#section=Related-Compounds-with-Annotation). In addition, the search on CAS and Reaxys gave no result. Although the suggested chemical names are quite correct, the chemical names of the compound in the manuscript are in accordance with the latest version of IUPAC rules.

  • All NMR spectra should be presented in the Supplementary Materials.

Author's Response: The 1H and 13C NMR spectra were added in the supplementary Materials.

  • Minor observations:As the authors have written "in silico" with Latin characters, "in vitro" should also be written using the same characters.

Author's Response: Corrected as suggested.

Round 2

Reviewer 3 Report

Comments and Suggestions for Authors

I appreciate  that the authors made the suggested corrections and I consider that the article can be published in the present form.